# Visual Assessment and Longitudinal Strain During Dobutamine Stress Echocardiography

**DOI:** 10.3390/diagnostics15121473

**Published:** 2025-06-10

**Authors:** Dijana Trninić, Jelena Jovanic, Aleksandar Lazarevic, Miron Marjanovic, Sinisa Kovacevic, Neno Dobrijevic, Snjezana Pejicic Popovic

**Affiliations:** 1Department of Cardiology, University Clinical Center of the Republic of Srpska, 78000 Banja Luka, Bosnia and Herzegovina; jelena.jovanic@kc-bl.com (J.J.); aleksandarlazarevicbl@gmail.com (A.L.); miron.marjanovic@kc-bl.com (M.M.); sinisa.kovacevic@kc-bl.com (S.K.); neno.dobrijevic@kc-bl.com (N.D.); spopovicpejicic@gmail.com (S.P.P.); 2Faculty of Medicine, University of Banja Luka, 78000 Banja Luka, Bosnia and Herzegovina

**Keywords:** dobutamine stress echocardiography, diagnostic study, longitudinal strain, viability study, visual assessment

## Abstract

**Background/Objectives**: Dobutamine stress echocardiography (DSE) is a non-invasive diagnostic technique commonly employed in routine clinical practice to identify coronary artery disease. Emerging echocardiographic methods, including strain and strain rate imaging, quantify alterations in myocardial contractility and may improve the diagnostic accuracy of DSE. The main aim of this study was to assess the correlation between visual interpretation and longitudinal strain during dobutamine stress echocardiography. **Methods**: Our study was observational and was conducted at the Cardiology Clinic of the University Clinical Center of the Republic of Srpska. It included 70 patients who underwent dobutamine stress echocardiography. The patients were divided into two groups (diagnostic and viable study). A visual assessment of segmental contractility of the left ventricle was performed, as well as an assessment of contractility with longitudinal strain (LS) during the test. **Results**: The median baseline LS of segments without impaired contractility in the diagnostic study was −20% (−21 to −18) and, at the peak of the test, −22% (−23 to −21), which was statistically significant (*p* < 0.05). The median baseline LS in the segments with impaired contractility was −17% (−18 to −16) and, at the peak of the test, −13% (−15 to −12), which was statistically significant (*p* < 0.05). In the viability study, the average baseline LS in the segments with improved contractility was −8% (−11 to −7) and, at the peak of the test, −14% (−17 to −13), which was statistically significant (*p* < 0.05). The average baseline LS in the segments without improved contractility was −6% (−5 to −7) and, at the peak of the test, −2% (−3 to −0), which was statistically significant (*p* < 0.05). **Conclusions**: Our study indicates a good correlation between a visual assessment of left ventricular segment contractility and longitudinal strain during dobutamine stress echocardiography.

## 1. Introduction

One of the most commonly used and accurate functional imaging tests for diagnosing ischemic heart disease is stress echocardiography, which includes dobutamine stress echocardiography [1,2]. The EVAREST observational multi-center study showed that overall sensitivity for all three types stress echocardiography including DSE was 95.4%, with a specificity of 96.0%. The positive predictive value and negative predictive value were 82.8% and 99.0%, respectively [2]. A powerful imaging method to diagnose cardiac disease is myocardial strain. Myocardial strain measures cardiac deformation, myocardial shortening in the longitudinal and circumferential directions, and thickening in the radial direction. It can be applied to any cardiac chamber. The most widely used clinical strain parameter is left ventricular (LV) global longitudinal strain by speckle-tracking echocardiography [3].

Newer echocardiographic techniques based on strain and strain rate have enabled the quantitative assessment of contractile function in individual ventricular segments, using clearly visualized “polar maps” that support differential diagnosis and therapeutic decision-making [4]. This advanced method is being increasingly integrated into stress echocardiography, demonstrating favorable results in predicting coronary artery disease, clinical outcomes, left ventricular remodeling, and both the presence and extent of ischemia [4]. The automated function imaging (AFI) technique can be applied during DSE, providing a reliable and nearly comprehensive evaluation of myocardial deformation throughout all stages of the stress protocol. When used with DSE, AFI has demonstrated feasibility rates ranging from 77 to 100 percent. Quantitative analysis methods can be incorporated into stress echocardiography on a routine basis [5].

Ischemia causes a prolonged time to the onset of systolic shortening (t-S), known as tardokinesis; decreased systolic strain and strain rates; and post-systolic shortening, which refers to additional shortening during isovolumic relaxation [6,7,8]. The utilization of speckle-tracking echocardiography enhances the sensitivity and accuracy of dobutamine stress echocardiography. Combining GLS cutoff with DSE resulted in a higher AUC than either method alone (0.9, *p* < 0.001), with 95.9% sensitivity, 84.6% specificity, 85.5% PPV, 95.7% NPV, and 90% diagnostic accuracy [9]. Longitudinal strain during DSE can also accurately assess myocardial viability [10].

Normal GLS values are derived from eight to 25 studies, encompassing 2396 patients with a mean age of 42 years and a body surface area of 1.7 ± 0.2 m^2^. The average GLS values are −21 ± 2.6%. This range is comparable to vendor reference values (General Electric—20 ± 2.4%, Philips—20.1 ± 2.4%, Siemens—20.0 ± 2.7%). Global longitudinal strain reference values may vary according to age over 60 years, gender, weight, and blood pressure. The absolute GLS difference between men and women exceeds 1% [11].

The objectives of this study were: (1) to examine the correlation between visual assessment and longitudinal strain during the DSE; (2) to compare the longitudinal strain (LS) values of the normokinetic, hypokinetic, and akinetic segments of the left ventricle at rest and the peak of DSE in diagnostic study; (3) to compare the longitudinal strain (LS) values of the normokinetic, hypokinetic, and akinetic segments of the left ventricle at rest and the peak of DSE in viability study; and (4) to examine changes in the longitudinal strain of the left ventricular segments during the diagnostic and viability studies.

## 2. Materials and Methods

Our study was observational and included 70 patients who underwent dobutamine stress echocardiography at the Cardiology Clinic of the University Clinical Center of the Republic of Srpska. The study was conducted from 1 January 2023 to 1 January 2024.

Ethical approval was obtained from the Ethics Committee of the University Clinical Center of the Republic of Srpska to ensure compliance with the ethical standards and research protocols. The study adhered to the ethical principles outlined in the Declaration of Helsinki, focusing on the protection of patients’ rights, privacy, and confidentiality throughout the study period.

One group of patients underwent DSE for ischemic disease (38 patients), and the other group for a viability test (32 patients) with a previous coronary event. Seventeen patients were excluded from the study due to poor echocardiographic image quality at the peak of the test.

The study inclusion criteria for the diagnostic study were as follows: suspected coronary heart disease, typical anginal pain, atypical anginal pain, risk factors for cardiovascular diseases, left bundle branch block and chest pain, previous coronary event and recurrent chest pain, and an inconclusive ergometric test.

The indication for the viability test was the assessment of myocardial viability after a coronary event to perform revascularization.

The exclusion criteria were chest pain of unstable angina type or myocardial infarction, arrhythmic instability of the patient, cardiac decompensation, severe valvular heart disease, and poor acoustic window.

All patients underwent DSE using a standard protocol with an incremental dobutamine infusion rate of 5, 10, 20, 30, and 40 µ/kg/min every 3 min, and up to 1 mg of atropine if the target heart rate (85% of the age-predicted maximum heart rate) was not achieved. Heart rate, blood pressure, 12-lead electrocardiography, and symptoms during DSE were recorded at each DSE stage. Betab- blockers and calcium channel blockers (non-dihydropyridines) were discontinued at least two days before the test. The criteria for terminating the test for ischemia were the completion of the protocol, development of new wall motion abnormality (WMA), severe chest pain, systolic blood pressure (SBP) > 220 mmHg or diastolic blood pressure (DBP) > 120 mmHg, symptomatic hypotension, and serious ventricular or supraventricular arrhythmias. The examinations were performed in the left supine position with a Vivid E9 scanner (GE Ultrasound). Complete two-dimensional, color, pulsed, and continuous wave Doppler echocardiography was performed at rest according to standard techniques [11]. Two cine loops from apical 4-, 3-, and 2-chamber views were recorded. All images were digitally stored on hard discs for offline analysis, and the wall motion was evaluated using the 17-segment model. Incremental low-dose dobutamine infusion at a maximal rate of 20 µg/kg/min was used in the assessment of the viability. Segmental wall motion at rest was scored on a four-point scale: (1) normokinetic (normal kinetics); (2) hypokinetic (decreased endocardial excursion and systolic wall thickening); (3) akinetic (absence of endocardial excursion and systolic wall thickening); and (4) dyskinetic or aneurysmal (paradoxical outward movement in systole) [5]. Contrast agents were not used in this case.

The calculation of the LS parameters was done using the EchoPAC (Vivid 9) workstation with the automated function imaging method.

We assessed the contractility of the left ventricular segment visually and by applying longitudinal strain before the test and at the peak of the test in diagnostic and viability studies. The DSE was performed by an experienced echocardiographer. Strain analysis was done on record. Visual analysis of wall movement was not performed independently of the strain analysis—i.e., blinded.

### Statistics

Results were presented as frequency (percent), median (IQR), and mean ± standard deviation (SD). Statistical hypotheses were tested using the Mann-Whitney test, Wilcoxon test, and Kruskal–Wallis test. All *p*-values less than 0.05 were considered significant. Statistical data analysis was performed using IBM SPSS Statistics 22 (IBM Corporation, Armonk, NY, USA).

## 3. Results

Our study included 70 patients who were divided into two groups. Thirty-eight patients were referred for the diagnostic study, and 32 patients for the viability study. The average age of the patients was 65.46 ± 7.58 years. The study included 32% diabetics, 92% patients with arterial hypertension, 74% with hyperlipidemia, and 18% smokers. The average BMI was 24.78 kgm^2^. The average EF was 51.68%,and the average LS value was −15.23%. A total of 1190 left ventricular segments at rest and peak of DSE were analyzed visually and using longitudinal strain. Table 1 presents a comprehensive demographic summary of the study population.

The median value of LS in normokinetic segments was −20% (−21 to −18), in hypokinetic segments −14% (−15 to −12), and in akinetic segments−6% (−7 to −5). The median value of the longitudinal strain at the peak of the test was −22% (−23 to −21) in normokinetic segments, −13% (−14 to −12) in hypokinetic segments, and −4% (−5 to −3) in akinetic segments. These results are presented in Figure 1.

The diagnostic study analyzed the value of the longitudinal strain of the segments visually assessed as segments with impaired and without impaired contractility. The average baseline LS of segments without impaired contractility in the diagnostic study was −20% (−21 to −18) and, at the peak of the test, −22% (−23 to −21). This difference was statistically significant (*p* < 0.05). The average baseline LS in the segments with impaired segments was−17% (−15 to −12) and, at the peak of the test, −13% (−15 to −12). This difference was statistically significant (*p* < 0.05). At the peak of the test, the segments with impaired contractility had a smaller median longitudinal strain than those without impaired contractility. During the test, segments with impaired contractility showed a greater change of −5% (−6 to −4) in the median longitudinal strain than segments without impaired contractility, which showed −2% (−1 to −3). Table 2 summarizes the results of the diagnostic study.

The viability study analyzed segments with and without improved contractility. The average baseline LS in the segments with improved contractility was −8% (−11 to −7) and, at the peak of the test, −14% (−15 to −3). This difference was statistically significant (*p* < 0.05). The average baseline LS in the segments without improved contractility was −6% (−8 to −5) and, at the peak of the test, −4% (−5 to −3). The change in the median longitudinal strain was significantly higher [−6% (−5 to −7)] than that in the segments without improved contractility [−2% (−3 to 0) (*p* < 0.05)]. Table 3 presents these results.

There is a strong positive and statistically significant correlation between visual assessment and longitudinal strain et the baseline DSE, proven with Person’s correlation coefficient < 0.01 (r_s_ = 0.866, *n* = 831, *p* < 0.01). Also, at the peak of the DSE between the value of longitudinal strain and visual assessment, there is a moderate, positive, and statistically significant correlation for *p*-values < 0.01 (r_s_ = 0.485, *n* = 824, *p* < 0.01).

## 4. Discussion

A small number of studies have examined the relationship between the visual assessment of the contractility of the left ventricular wall segments and the longitudinal strain during dobutamine stress echocardiography.

In our study, we recorded a decrease in the values of the longitudinal strain parameters of the left ventricular segments, which were described as normokinetic and hypokinetic, during DSE.

Left ventricular segments that were visually described as hypokinetic and akinetic had lower longitudinal strain values both at rest and peak during dobutamine stress echocardiography.

Similar findings were reported by Karolina et al., who compared visual assessment with longitudinal strain during DSE. In their study, the longitudinal strain of normokinetic segments was −16.33% ± 6.26 at rest and −15.37% ± 6.89 at peak. For hypokinetic segments, the longitudinal strain measured −13.05% ± 6.89 at rest and −11.37% ± 5.11 at peak. Akinetic segments showed longitudinal strain of −8.46% ± 5.69 at rest and −7.37% ± 3.92 at peak [12].

In our DSE study on ischemia, segments with impaired contractility showed a reduction in longitudinal strain at peak stress compared to baseline. Median peak LS was lower in segments with impaired contractility than in those with preserved function. Comparable results were reported by Karolina et al. The baseline longitudinal strain in segments with impaired contractility was −16% (−14 to −18), compared to −17% (−14 to −20) in segments without impairment, *p* = 0.323. At peak stress, longitudinal strain in impaired segments was −9% (−16 to −13), whereas in segments without impairment, it was −16% (−11 to −21), *p* < 0.001 [12].

Wierzbowska-Drabik et al. demonstrated a significant decrease in longitudinal strain at the peak of DSE, with a more pronounced reduction observed in segments showing impaired kinetics. Peak systolic longitudinal strain, assessed using standard speckle tracking echocardiography (STE) and automated function imaging (AFI), as well as strain rate, differed between normokinetic and hypokinetic/akinetic left ventricular segments. These differences were evident both at rest and at peak stress, with normokinetic segments consistently showing the highest absolute values of systolic longitudinal strain (SLS) and systolic longitudinal strain rate (SLSR). The drop in absolute SLS from baseline to peak stress was more substantial in segments with impaired contractility than in those with a normal stress response. In normokinetic segments, baseline SLS was −16.5 ± 5.9%, decreasing to −15.9 ± 7.5% at peak. Hypokinetic segments had a baseline SLS of −15 ± 5.7%, while akinetic segments showed −12.7 ± 6.7%. At peak stress, SLS in hypokinetic and akinetic segments was −14.4 ± 7.4% and −13.7 ± 8.9%, respectively. In segments with contractile dysfunction during DSE, a significant reduction in absolute SLS was observed, from −16.6% to −14.1%, *p* < 0.0011 for SLS-AFI [13].

Voigt et al. investigated strain rate imaging (SRI) parameters indicative of stress-induced ischemia and evaluated their clinical utility. The study enrolled 44 patients with confirmed or suspected coronary artery disease. Regional ischemia was defined using simultaneous perfusion scintigraphy as the reference standard. Coronary angiography was performed in all participants. Segmental strain and strain rate were assessed across all stages of stress by quantifying deformation amplitude and timing, as well as through visual curved M-mode analysis. Peak systolic strain rate increased significantly with dobutamine stress (−1.6 ± 0.6 s vs. −3.4 ± 1.4 s, *p* < 0.01) in nonischemic segments, while it was reduced in ischemic segments (−16% ± 7% vs. −10% ± 8%, *p* < 0.05) [14].

When a single lesion was considered, peak stress LS and the left anterior descending artery (LAD)regional longitudinal strain (RLS) were lower in the obstructed LAD regions than in normoperfused territories (17.4 ± 5.5 vs. 20.5 ± 4.4%, *p* = 0.03; 17.1 ± 7.6 vs. 21.6 ± 5.5%). The addition of RLS to regional WMSI was able to improve the accuracy of LAD significant coronary stenosis (SCS) prediction (AUC 0.68, *p*  =  0.037). DSE strain analysis is feasible and may improve the prediction of LAD SCS, whereas regional WMSI assessment performs better in the presence of SCS of the circumflex artery (LCX) and the right coronary artery (RCA) [15].

In the group of patients who underwent DSE for viability, there was an increase in the longitudinal strain in the segments in which there was an improvement in the contractility of the left ventricular segments. Ismail et al. investigated myocardial viability in patients with ST-segment-elevation myocardial infarction after fibrinolytic therapy with low-dose DSE. The dobutamine-induced strain and strain rate were significantly higher in viable segments than in non-viable ones (10 out of 16 for S and 11 out of 16 for SR). A cutoff value ranging from −8.5 to −9.6% for the S identified viability in apical and mid-segments, whereas a cutoff value ranging from −11.5 to −21.5% identified viability in basal segments [16]. In our study, LS was also higher in left ventricular segments with preserved viability.

The results of the study by Sharma et al. confirm that strain and strain rate values increase in viable segments during dobutamine stress echocardiography. Research has shown that peak longitudinal strain rate [AUC 0.83 (95% confidence interval (CI) 0.67–0.99), *p* = 0.001; optimal cutoff—0.64 s^−1^ for sensitivity 0.87 and specificity 0.81], post-dobutamine peak longitudinal strain rate [AUC 0.94 (95% CI 0.87–1.00), *p* = 0.001; optimal cutoff—0.85 s^−1^ for sensitivity 0.89 and specificity 0.77], and change in peak longitudinal strain rate [AUC 0.93 (95 CI 0.86 s^−1^
*p* = 0.001; optimal cutoff—0.2 s^−1^ for sensitivity 0.87 and specificity 0.87] predicted viability [17].

Hanafy et al. also confirmed in a study that included 60 subjects after myocardial infarction that dobutamine-induced peak longitudinal strain was greater in viable than in non-viable segments (<0.001 for mid-inferoseptum, *p* = 0.001 for mid-inferolateral, and <0.001 for all other segments) [18].

Bhutani and colleagues examined the feasibility of applying stress in the viability test during the dobutamine stress echocardiographic test in postinfarction patients. The results of the study showed that the longitudinal strain values were significantly higher in viable segments (−9.4901%) than in non-viable ones (−3.4847%) (*p* < 0.01). There was also a significant difference in strain rate values between viable (−1.0309%) and non-viable segments (−0.4827%) (*p* < 0.01) [19].

Using the Pearson`s correlation coefficient, we determined a statistically significant correlation between the value of longitudinal strain and visual assessment and the rest and peak dobutamine stress test. Saleh et al. found that global longitudinal strain at stress moderately correlated with the presence of significant coronary artery disease (CAD), r = 0.41, *p* < 0.0001, and had a moderate correlation with the severity of coronary occlusion, r = 0.62, *p* < 0.0001 [20].

Although speckle-tracking echocardiography provides quantitative data describing the contractile function of each ventricular segment and greater accuracy of segment assessment, visual assessment of contractility remains the gold standard. Speckle-tracking echocardiography improves the accuracy of dobutamine stress echocardiography [12]. The novel AFI method appears faster, offering user-friendly polar maps of the left ventricle and, thus, a potential to become a useful clinical tool outside the research setting [21].

## 5. Conclusions

Our study indicated a good correlation between a visual assessment of left ventricular segment contractility and longitudinal strain during dobutamine stress echocardiography. Longitudinal strain values were different in normokinetic, hypokinetic, and akinetic segments at rest and the peak of the test. Changes in longitudinal strain values were greater in segments with impaired contractility in the diagnostic and viability studies. The application of longitudinal strain during dobutamine stress echocardiography increases the accuracy of these diagnostic methods. This imaging modality should be used in daily practice because it is available, reproducible, accurate, radiation- and contrast-free, and inexpensive.

## Figures and Tables

**Figure 1 diagnostics-15-01473-f001:**
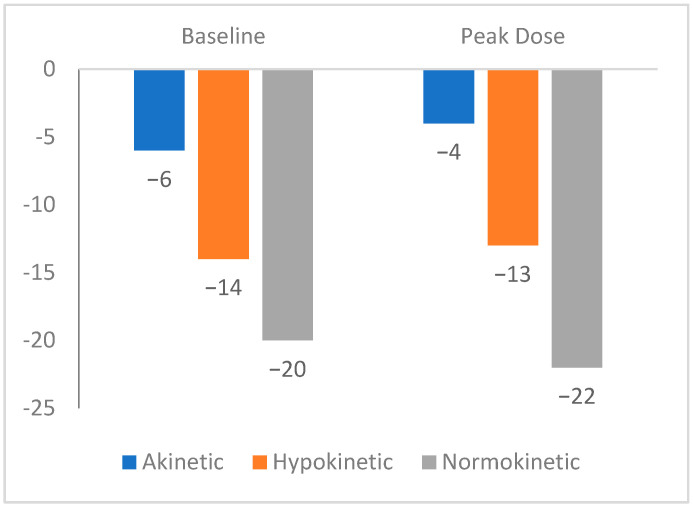
Longitudinal strain and visually assessed contractility.

**Table 1 diagnostics-15-01473-t001:** Clinical and echocardiography characteristics of the study population.

Parameter	Value
Clinical characteristics	
Age(years)	65.46 ± 7.58
BMI (kg/m^2^)	24.78 ± 1.93
HTA, *n* (%)	46 (92%)
DM, *n* (%)	16 (32%)
Smoking, *n* (%)	9 (18%)
Dyslipidemia, *n* (%)	12 (26%)
Echocardiographic parameters	
LVEF baseline (%)	51.68 ± 12.48
GLS baseline (%)	−15.23 ± 9.95

Values are *n* (%) or mean ± SD, SD = Standard deviation, BMI = Body mass index, HTA = Arterial hypertension, DM: Diabetes mellitus, LVEF = Left ventricular ejection fraction, GLS = Global longitudinal strain.

**Table 2 diagnostics-15-01473-t002:** Diagnostic study: comparison of longitudinal strain in baseline and peak dobutamine stress echocardiography.

Parameter	Segments with Impaired Contractility (*n* = 208)	Segments Without Impaired Contractility (*n* = 232)	*p*
Baseline longitudinal strain (%)	−17 (−18 to −17)	−20 (−21 to −18)	<0.05
Peak longitudinal strain (%)	−13 (−15 to −12)	−22 (−23 to −21)	<0.05
Changes longitudinal strain (%)	−5 (−6 to −4)	−2 (−1 to−3)	<0.05

Values are median (IQR). IQR = interquartile range.

**Table 3 diagnostics-15-01473-t003:** Viability study: comparison of longitudinal strain in baseline and peak dobutamine stress echocardiography.

Parameter	Segments with Improved Contractility (*n* = 208)	Segments Without Improved Contractility (*n* = 232)	*p*
Baseline longitudinal strain (%)	−8 (−11 to −7)	−6 (−8 to −5)	<0.05
Peak longitudinal strain(%)	−14 (−17 to −13)	−4 (−5 to −3)	<0.05
Changes longitudinal strain (%)	−6 (−5 to −7)	−2 (−3 to 0)	<0.05

Values are median (IQR). IQR = interquartile range.

## Data Availability

The data presented in this study are available on request from the corresponding author.

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
