# Peer review of "Visual Assessment and Longitudinal Strain During Dobutamine Stress Echocardiography"

_diagnostics, 2025, doi:10.3390/diagnostics15121473_

Round 1
Reviewer 1 Report
Comments and Suggestions for Authors
An interesting study showing the correlation between visual assessment of contractility and strain measurements. I have 3 main questions: 1. Were patients rejected from the study if accurate strain measurements could not be obtained? Surely not all patients was strain able to be analyzed at rest and peak? 2. Was the visual analysis of wall motion performed independently of strain analysis - ie blinded? 3. Why do they conclude that "The application of longitudinal strain during dobutamine stress echocardiography increases the accuracy of these diagnostic method"?
Author Response
Dear Reviewer, please check our reply in the attachment.

Reviewer 2 Report
Comments and Suggestions for Authors
-
Although the study results are good, strain echo is not a practical application.
-
I wonder that did you do interobserver (kappa statistics ?) and intra-observer (correlation coeeficent?) reliability test?
-
How were the anlaysis done ? on-record or off-record?
-
Is there statistical evidence of a good correlation between LS and DSE?
-
How much can it change our daily practice?
-
Both introduction and discussion in main text is too long.
- The English should be improved.
-
Although the study results are good, strain echo is not a practical application.
-
I wonder that did you do interobserver (kappa statistics ?) and intra-observer (correlation coeeficent?) reliability test?
-
How were the anlaysis done ? on-record or off-record?
-
Is there statistical evidence of a good correlation between LS and DSE?
-
How much can it change our daily practice?
-
Both introduction and discussion in main text is too long.
- The English should be improved.
Author Response
Dear Reviewer 2, please check our reply in the attachment

Round 2
Reviewer 2 Report
Comments and Suggestions for Authors
Thank you for this revision